# Swine Practitioner Practices on Oral Fluid Sampling in U.S. Swine Farms: A Nationwide Survey

**DOI:** 10.3390/pathogens14090940

**Published:** 2025-09-16

**Authors:** Xiaomei Yue, Mickey Leonard, Margret Tavai-Tuisalo’o, Sasidhar Malladi, Mariana Kikuti, Claudio Marcello Melini, Pam Zaabel, Marie R. Culhane, Cesar A. Corzo

**Affiliations:** 1Department of Veterinary Population Medicine, University of Minnesota, Saint Paul, MN 55108, USA; yue00075@umn.edu (X.Y.); mkikuti@umn.edu (M.K.); melin145@umn.edu (C.M.M.); grame003@umn.edu (M.R.C.); 2Department of Veterinary and Biomedical Sciences, University of Minnesota, Saint Paul, MN 55108, USA; leona314@umn.edu (M.L.); tauga001@umn.edu (M.T.-T.); sasidhar.malladi@gmail.com (S.M.); 3National Pork Board, Des Moines, IA 50325, USA; zaabel@aasv.org

**Keywords:** oral fluid, sampling procedure, disease surveillance, swine farms, swine disease

## Abstract

Oral fluid sampling has been widely adopted in swine health surveillance, offering an effective and cost-effective method for monitoring endemic and emerging diseases. This study characterized oral fluid sampling practices within U.S. pig production systems. An online questionnaire was conducted between June and October 2023, targeting swine practitioners, to collect data on implementation, usage, sampling protocols, and handling procedures. A total of 67 valid responses were received, representing an estimated 58M growing pigs and 3.9M sows. Nearly all respondents (99%) reported being familiar with or using oral fluid sampling for diagnostic purposes. The median of ropes hung per barn was two (interquartile range [IQR]: 1; 4), with 68% using one rope per two pens. The median of pens sampled per barn was six (IQR: 4; 10). Pigs typically accessed the rope for a median of 20 min (IQR: 17.5; 30). Sampling frequency varied by farm types. Half of gilt-development-unit (GDU) respondents collected samples monthly. When submitting samples to the veterinary diagnostic laboratories, pig age (91%) was the most frequently included information. This study reveals the widespread and varied adoption of oral fluid sampling, highlighting the need for standardized collection procedures to support consistent interpretation and improve reliability for detecting emerging pathogens.

## 1. Introduction

Oral fluids are obtained from pigs by allowing them to chew on a suspended absorbent cotton rope for 15–30 min, the time in which oral fluids are absorbed by the rope. Oral fluids are extracted by squeezing the rope and collecting the fluids into an sterile container which will then be submitted to the diagnostic laboratory for either pathogen or antibody testing [1,2]. There has been widespread use of oral fluids as a sample for surveillance, monitoring, and disease detection in the United States (U.S.) swine industry for at least a decade [1,2,3]. Several advantages that have led to its widespread adoption in the industry are reported to be its accuracy, effectiveness, and cost-efficiency for detection; the fact that it is non-invasive and welfare-friendly for pigs; and the fact that it is simple and safe for caretakers to collect [1,2].

In the U.S., veterinary diagnostic laboratories (VDLs) started offering oral fluid-based tests in 2010, with a subsequent rapid increase in usage. Bjustrom-Kraft et al. reported a total of nearly 370,000 tests conducted on oral fluid specimens between 2010 and 2016 in three major swine-focused VDLs [4]. Different studies have been conducted to assess the sensitivity of this sample type for common swine pathogens present on U.S. pig farms, such as porcine reproductive and respiratory syndrome virus (PRRSV), porcine epidemic diarrhea virus (PEDV), influenza A virus (IAV), porcine delta coronavirus (PDCoV), and porcine circovirus type 2 (PCV2) [5,6,7,8]. Specifically for PRRSV diagnosis, oral fluids comprise about 20% of monthly submissions from 2015 to 2020 [9]. Even though oral fluid is a widely accepted sampling methodology, it is currently unknown how widespread its use is within and across production systems or veterinary clinics. Oral fluids can be collected from both individual pigs and pig pens; however, this study focused specifically on pen-level, population-based sampling, as its implementation is more variable and warrants further characterization.

Many studies have reported their standardized oral fluid sample collection, suggesting the use of a three-stranded cotton rope in the pen for 20 to 30 min, placing the wet portion in a clean plastic bag, manually extracting the fluid, decanting it into a tube, and refrigerating the samples before laboratory submission [2,5,10,11]. However, there are notable differences in the specific steps for collecting oral fluids, both in experimental conditions and in actual production practice. One rope per pen provided effective detection of PRRS and IAV in previous studies for pens with fewer than 30 pigs [12,13,14]. However, in practice, hanging one rope between two pens is a common sampling method, but detection results may be impacted when one of the pens is free of a specific pathogen [15]. The rope was generally reported to remain hung in pens for 20 to 30 min [4,16], although some studies also reported shorter (e.g., 15 min [17]) or longer durations (e.g., 45–60 min [18,19]). Tarasiuk et al. reported that increasing the number of ropes available in a pen and extending the sampling duration can enhance pig participation in pen-based oral fluid sampling [20]. Although oral fluid sampling methodologies have been documented in several studies, information regarding sampling practices, sample management, and whether oral fluid is used for active or passive surveillance programs in practice remains unknown.

Foreign animal disease (FAD) emergence continues to be a major concern in the swine industry, with African swine fever (ASF) being a serious threat in the U.S., necessitating effective surveillance strategies for early detection and response [21]. In this context, the United States Department of Agriculture-Animal and Plant Health Inspection Service-Veterinary Services (USDA-APHIS-VS) is evaluating the inclusion of oral fluid sampling in the ASF surveillance and response policies [22]. Oral fluid serves as an alternative diagnostic specimen, minimizing contact with blood from infected animals, which has been shown as a significant risk for further transmission of ASF [23,24]. Mur et al. (2013) have demonstrated the promise of using oral fluid for ASF diagnosis and efficient surveillance [25], yet the feasibility needs to be further investigated, as the industry’s uptake of oral fluid sampling remains unassessed. Therefore, information regarding how pig production systems and practitioners are using this sampling methodology is important to better understand how this sampling methodology aids in the early detection of emerging pathogens.

This study set out to characterize the current use of population-based oral fluid sampling in pig production systems in the U.S. swine industry. Specifically, this study aimed to (1) gather insights from the target population, including veterinarians and producers in the U.S. swine industry; (2) determine whether oral fluids are being implemented and what their primary use is; (3) characterize current sampling (i.e., frequency, sample size among others) protocols; and (4) characterize sample handling/processing and submission protocols. The findings from this survey can inform the diversity of the sampling practices and underscore the need for standardization, contributing to improved oral fluid sampling strategies both in the U.S. and globally.

## 2. Materials and Methods

The University of Minnesota Institutional Review Board (IRB) approved this study and classified it as one that is not research involving human subjects (IRB ID: STUDY00020075). Contact information was collected solely to assess response relevance and eligibility, as well as for follow-up communication (e.g., gift card distribution), but was deleted prior to analysis, and not used in any part of the data analysis.

### 2.1. Questionnaire Development

An online questionnaire was developed using Qualtrics XM software (Copyright © 2023 Qualtrics), in which a variety of multiple-choice and open-ended questions were included. The questionnaire gathered information in four main sections aligned with the objectives of this study: (1) respondent demographics, (2) oral fluid implementation and primary use, (3) oral fluid sampling protocols, and (4) sample handling/processing and submission. The developed survey was initially sent to a convenient sample of pig production companies as a pilot survey (see Appendix A). Three production systems, representing approximately 200 K sows and their associated growing pig production, were approached for the pilot survey. This pilot phase aimed to address any overlooked design issues, validate the questions, and identify necessary adjustments to improve clarity and completeness [26]. The final questionnaire was refined based on the responses from the pilot survey. Questions related to site-level details managed by the respondent (e.g., number of sites, barns, pigs per barn or pen) were removed, as collecting such information from individuals, like the director of health responsible for dozens or even hundreds of farms, was impractical and less relevant to the characterization objective of this study. Additionally, a few questions were revised for clarity; the question “familiar with oral fluid collection and its use for diagnostics (Yes/No)” was changed to “use oral fluid for diagnostics (Yes/No)”; the question on sampling frequency was expanded to specify sampling by farm type (e.g., gilt-development unit (GDU), sow farm, growing pig farm); the question on “number of ropes per barn when collecting oral fluids” was split into two questions asking the number of ropes per barn and pigs when collecting oral fluids for active surveillance and clinical signs. The final questionnaire included 26 questions, summarized in Table 1, with the full content available in Appendix A. The questionnaire was accompanied by a cover letter stating the available languages (English and Spanish), the objectives of the survey, and the confidentiality of responses emphasizing that answers would be aggregated and shared in a summary report without personal or company identifiers. The letter also informed participants of a USD 25 gift card compensation for completing the survey.

### 2.2. Questionnaire Distribution

The target population of this study included U.S. swine practitioners with knowledge of oral fluid usage, primarily veterinarians, producers, and farm personnel, who could provide informed insights into whether and how oral fluids are used in swine health management. To reach this population, the survey was distributed through two major channels: (1) direct email contact with producers, veterinarians, and vet clinics participating in the Morrison Swine Health Monitoring Project (MSHMP), the largest voluntary swine health monitoring project in the U.S., covering approximately 60% of the national sow population [27,28]; and (2) inviting U.S.-based American Association of Swine Veterinarians (AASV) members via the AASV weekly e-letter, which reaches a broad national network of swine health professionals. The questionnaire link was accessible via email from 7 June 2023, to 23 October 2023, and it was included in five AASV weekly e-letters distributed from 13 September 2023, and closed on 23 October 2023.

### 2.3. Data Cleaning and Analysis

The survey data collected from the online questionnaire was consolidated into a Microsoft Excel (Version 16.100.3, Redmond, Washington, U.S.) spreadsheet. The dataset underwent two rounds of thorough manual cleaning to ensure the accuracy and relevance of the responses. In the first round, two individuals jointly reviewed all submissions to minimize manual errors and assess the credibility of each entry. This joint review helped reduce the risk of oversight or misclassification. Each response was evaluated for internal consistency and plausibility, such as whether the name, contact information, affiliated company or clinic, and reported pig inventory aligned logically. Responses were excluded if they contained clearly fabricated or incomplete information, such as unrealistic names (e.g., “121121,” “A:,” or “dfgs dszf”) or invalid company or clinic names (e.g., “sde,” “NA,” “no,” or “ppp”). Entries were also removed if they exhibited implausible or uniform responses across the questionnaire, such as filling in “1,” “no,” or “NA” for most or all questions. Additional eligibility screening was conducted to exclude responses from outside the U.S., or those representing non-commercial operations (only overseeing a small number of pigs or 0 pig). In the second round, a third individual independently reviewed the dataset that had been cleaned in the first round to further identify and remove any remaining suspicious or potentially fraudulent entries. In total, 385 responses were excluded during this process. Notably, the majority of invalid responses were a result of the publicly accessible survey link being shared on websites that compile surveys offering gift card incentives, which attracted a large number of irrelevant or ineligible participants. To address this, the cleaning process was guided by the principle of retaining only eligible and credible responses, to ensure an accurate and representative characterization of oral fluid usage in the U.S. swine industry.

Descriptive statistics were then performed on the cleaned data, including calculating the mean, median, and interquartile range (IQR) for quantifiable survey questions. Specifically, in question Q8, the respondents were asked to summarize the sampling frequency for active surveillance according to the farm type (i.e., gilt-development units (GDU), sow farm, and growing pig farm). Responses were organized and processed as follows: (a) answers were summarized according to what was actually provided, since some respondents only reported on the farm types they managed, rather than all three; and (b) when the same farm type applied multiple frequency (e.g., a GDU sampled both weekly and monthly), the response was counted as multiple entries, as it likely reflected practices across different GDUs. For open-ended questions, some responses included ranges instead of exact numbers. In such cases, we calculated the average of the upper and lower limits of the range. Answers that could not be converted into numbers were described separately. Additionally, categorical data was summarized using frequency counts and percentages to provide an overview of the distribution of responses.

## 3. Results

A total of 452 responses were initially collected, of which 67 valid responses remained after data cleaning, including 8 from the pilot survey phase. The 385 exclusions were due to ineligible or irrelevant entries. The final results were summarized according to the four sections of the questionnaire. We utilized the eight results from the pilot survey; as the majority of the content in the pilot survey and the final questionnaire overlapped, the few differing questions (i.e., frequency of sampling for active surveillance by farm type, frequency of sampling—no clinical signs, number of ropes per barn and pigs when collecting oral fluids for active surveillance or based on clinical signs) were analyzed only based on the final questionnaire responses.

### 3.1. Demographics

Among 67 valid respondents, 91% were veterinarians and 9% were farm personnel including 5 serviceman/service manager from nursery and finishing sites, and 1 gilt development manager. Both veterinarian and farm personnel roles were either responsible for overseeing herd health across multiple farms within a production system/clinic or had direct, hands-on experience in day-to-day farm operations. These roles enabled them to actively participate in sampling and to possess a solid understanding of sampling procedures. The respondents represented 42 companies or clinics, and collectively accounted for an estimated total of 58 million growing pigs and 3.9 million sows. While these figures may be inflated due to potential overlap in responses from affiliated entities, the coverage still reflects a substantial portion of the U.S. swine industry, which reported a national inventory of 75 million hogs and pigs and 6 million sows, as of 1 December 2023 [29]. The 42 companies and clinics cover the major U.S. pig-producing regions, including, but not limited to, the Midwest and South, reflecting the geographic representativeness of the survey results.

### 3.2. Oral Fluid Implementation and Primary Use

Almost all (66 out of 67) respondents stated that “they are familiar with oral fluids collection and its use for diagnostics” or “use oral fluids for diagnostics”, which indicates the widespread use of this sampling methodology for diagnostics in pig production systems.

In general, the primary use of oral fluids as a diagnostic sample was mostly (40/66, 61%) for active surveillance (regardless of clinical signs), followed by 33% (22/66) for passive surveillance based on the appearance of clinical signs (e.g., cough, sneeze, diarrhea, mortality), 3% (2/66) based on pig shipments, and 3% (2/66) for both active and passive surveillance.

### 3.3. Oral Fluid Sampling Protocols

#### 3.3.1. Sampling Frequency

The frequency of oral fluid collection for active surveillance is summarized in Figure 1. A total of 161 answers were received for this question Q8 for all three different farm types. Among them, 35% (56/161) collected oral fluids monthly, 27% (44/161) collected oral fluids weekly, 4% (7/161) collected bi-weekly and only 2% (4/161) collected oral fluids daily. Other frequencies such as quarterly, semi-annually, or as needed, account for 31% (50/161) of the answers.

Overall, oral fluid sampling frequency varied by site type (Figure 1). Among the three site types, the most frequent collection of oral fluids for active surveillance (e.g., daily, weekly, bi-weekly, monthly) was the GDU in which half of them collected oral fluids monthly, whereas 33% (20/60 GDUs) did it weekly. For sow farms, the most common oral fluid sampling frequencies were monthly (14 answers) and weekly (3 answers), with some farms also sampling as needed (7 answers) or based on animal movement such as multiplier/nucleus shipping piglets (3 answers). Less frequent practices included bi-weekly, semi-annually, and whenever there is an outbreak. For growing pig farms, the most reported frequencies were weekly (21 answers) and monthly (12 answers), with additional practices such as when clinical signs appear (6 answers) and sampling as needed, without additional information (6 answers). For herds that do not routinely sample oral fluids, some responses specified factors influencing the sampling frequency: animal movement and clinical signs. Animal movement was identified as a factor across three herd types; for instance, in GDU, oral fluid samples were collected “when gilts are shipped, upon arrival, and at end of quarantine”; or “2–4 days post-arrival and 4–5 days before entry into sow farms”; in sow farms, oral fluids were “used for post-arrival testing in isolation facilities (e.g., 2 and 7 days after arrival)” or “tested prior to gilts entering the sow herd”; and in growing pig farms, oral fluids were sampled in the middle of the production phase and prior to shipment. Meanwhile, the presence of clinical signs often triggers passive surveillance through sampling in growing pig farms, a topic further explored in subsequent questionnaire responses.

Oral fluid sampling frequency in the presence or absence of clinical signs varied (Figure 2). On the one hand, when clinical signs were present, half of the sites collected oral fluids more frequently on a weekly (36%, 24/66) or daily (14%, 9/66) basis. On the other hand, 26% (17/66) of sites collected oral fluid samples only once, for disease diagnosis, and found that there is little need to continue sampling and testing. When no clinical signs were present, active surveillance was primarily conducted on a monthly (28%, 16/58) or weekly (21%, 12/58) basis. Other factors accounted for 35% (20/58) of the responses, including farm type (16%, 9/58) and animal movement (19%, 11/58) (Figure 2). Furthermore, to understand how clinical signs influenced oral fluid sampling frequency in practice, we analyzed responses reporting regular monitoring (i.e., daily, weekly, monthly) for active surveillance and examined how these frequencies changed when clinical signs were present (Figure 3). Among the 31 responses indicating active surveillance through oral fluid sampling from the questionnaire, 64% (20/31) reported an increased sampling frequency (e.g., shifting from monthly to weekly, monthly to daily, or weekly to daily) when clinical signs appeared. In 23% (7/31) of cases, the sampling frequency remained unchanged, regardless of clinical signs, while 13% (4/31) of respondents chose to decrease the frequency or conduct a single sampling when clinical signs arose.

Of the 66 respondents who collected oral fluid samples, 65 agreed that sampling frequency is influenced by additional situations, while one indicated they maintained their weekly sampling approach, regardless. These 65 respondents provided 121 answers detailing other influencing situations. The two most frequently mentioned influencing situations were pre-shipping sampling (23%, 28/121) and on-arrival (post-placement) sampling (23%, 28/121). These were followed by active surveillance (16%, 19/121) and passive surveillance, due to a suspicion of infection with IAV, PRRS, or PED in the sites, due to the presence of clinical signs and the desire to obtain a diagnosis (16%, 19/121). Disease pressure in the surrounding area farms was also mentioned in 7% (8/121) of the listed concerns. The impact of disease pressure may vary, as one responded that PRRS in the area will designate more frequent testing while another respondent stated that sites in cleaner or low-dense regions will be tested more frequently. Other listed situations were the location of the sites nearby, pig movement (not specified), sow farm health status, current site status, and risk to other farms.

#### 3.3.2. Sample Size

For the number of ropes used per sampled pen, the majority (68%, 45/66) of respondents use one rope for every two pens, 21% (14/68) use one rope per sampled pen, while the remaining respondents employ alternative approaches, such as using two ropes per pen. At the barn level, the median number of ropes per barn was 2, with the IQR of 1 to 4 (Table 2). The median number of pens sampled per barn is 6, ranging between 1 and 30. Notably, five complex responses were not included in the table, three stated that they test all pens in the barn and hang one rope per pen without mentioning the total number of pens in the barn, and two answers only reported the number of pigs per rope in the barn without mentioning the number of ropes hung per barn. These answers indicated that the number of pens sampled per barn would vary, based on barn size and layout.

The term “airspace” was not used in our questionnaire, but it frequently appeared in responses to open-ended questions it could represent a room or a barn, depending on the layout of the farm. Adopting this term helped clarify the responses, as it accounted for the variations in barn design and site size among the respondents. During active surveillance, one rope was typically used for a median of 500 pigs in the airspace. This median remained the same when collecting oral fluids in response to clinical signs. However, the IQR of pigs per rope was slightly higher for passive surveillance triggered by clinical signs (288; 700) compared to active surveillance (263; 650). A further t-test comparing the number of pigs per rope between routine and clinical scenarios indicated no statistically significant difference (*p* = 0.55). Among the 66 responses, 39 provided quantifiable data on the number of pigs per rope for both active and passive surveillance. Of these 49% (19/39) did not change the number of ropes used, based on the presence of clinical signs, while 26% (10/39) increased the number and 26% (10/39) decreased it (not shown in Table 2). Additionally, the number of pigs represented by a single oral fluid sample had a median of 138, ranging from 5 to 1200. The large number of pigs represented by an oral fluid sample corresponds to the practice of aggregating samples of oral fluid between different pens, which was specifically mentioned in 11 responses. For example, one response specified that a single oral fluid sample represents 150 pigs, as they aggregated oral fluids from 3 ropes, each hanging across 2 pens of 25 pigs, resulting in 25 pigs per pen × 2 pens per rope × 3 ropes per sample.

When determining the number of ropes needed to be used, 54% (36/66) of the respondents decided by themselves, 29% (19/66) referred to the number dictated by company/clinic guidelines, and 9% (6/66) used the number dictated by their veterinarian; the remaining 8% (5/66) reported using a multiplication system guideline, while decisions for wean-to-market farms were made by the health director, based on other company guidelines, research, or the specific disease and site health status.

#### 3.3.3. Personnel Responsible for Oral Fluid Sample Collection

Among the 66 respondents using oral fluids for diagnosis, in 32 cases (48%) the caretaker collected the sample, 12 respondents (18%) said they were collected by the veterinarian or the site supervisor, 12 (18%) by the regional production supervisor, and 5 (8%) by the company/clinic health technician. The remaining (8%) respondents indicated that collection is carried out by the team, including all of the above roles, or as needed.

#### 3.3.4. Length of Time Pigs Have Access to the Rope

The median length of time pigs have access to the rope was 20 min (IQR: 17.5, 30) (Figure 4). One answer which reported hanging the rope for 24–48 h was removed from the analysis. A total of 54% of the responses reported hanging the rope for 20 to 30 min, while 25% hung it for less than, or equal to, 15 min. Two respondents also mentioned that the time pigs have access to the rope depends on age and size; for instance, finishing pigs have shorter access times, to prevent rope loss and ensure the ropes can be retrieved.

#### 3.3.5. Source and the Diameter of the Rope

The source of the rope was unknown for the majority (64%, 42/66) of the respondents. The known store/vendor varied among 18% (12/66) of the responses, and the online source accounted for 7% (5/66). The remaining 11% (7/66) of respondents stated that the rope was ordered through the company or clinic office. The rope diameter was unknown or unspecified in 44% (29/66) of responses, with the remaining most commonly used being 1/2 inch (11 responses), 5/8 inch (9 responses), 3/4 inch (7 responses), and 1 inch (5 responses).

### 3.4. Sample Handling, Processing and Submission

#### 3.4.1. Availability of Written Protocol for Handling/Processing the Ropes and Oral Fluids

The majority (71%, 47/66) of the respondents answered that they have a written protocol available for handling and processing ropes. A total of 50% (33/66) of the respondents indicated that farm personnel have a copy of the protocol, 6% (4/66) replied that farm personnel do not have a hard copy, while 15% (10/66) did not specify whether a hard copy is available for farm personnel. Meanwhile, 29% (19/66) of the respondents do not have a written protocol to guide the rope handling and processing.

#### 3.4.2. Storage of Collected Samples

Once oral fluid samples are collected, 76% (50/66) of the responses specified that samples were stored in refrigeration, of which half were stored (38%, 25/66) at the farm, and the other half (38%, 25/66) were left in the veterinarian or regional production supervisor truck under refrigeration, until they arrive at the office. A total of 8% (5/66) of responses indicated that the samples were placed on ice in a cooler and driven or shipped to the lab. Only 9% (6/66) of the respondents reported that samples would not be kept refrigerated until they arrived at the office for refrigeration. In 6% (4/66) of the cases, respondents stated that the logistics may vary based on the schedule of the day, without specifying the refrigeration situation. Additionally, one response reported that the samples are either placed on ice or in a refrigerator.

#### 3.4.3. The Time Range Between Collecting the Oral Fluid Sample and Submission to the Diagnostic Laboratory

Regarding the time between oral fluid sample collection and submission to the diagnostic laboratory, 91% (60/66) of respondents were aware of the time range, and 74% (49/66) reported that samples were delivered to the laboratory within an average of 24 h. However, there were variations in the minimum and maximum time ranges for sample submission (Table 3). Two of the respondents reported that these times would vary during weekends.

#### 3.4.4. Destination of Oral Fluid Samples upon Collection

After collection, the majority of the collected samples were first processed at the local production company lab or clinic (45%, 30/66) and then shipped to the VDL. In 35% of cases (23/66), samples were sent directly from the farm to the VDL. For 12% of responses (8/66), the route depended on the situation; samples were either shipped directly to the VDL or driven to their main office for mailing to the VDL. Additionally, 5% (3/66) of indicated samples were sent to the veterinarian’s clinic without specifying the next destination, while 3% (2/66) of reported samples were sent to the production company’s laboratory.

#### 3.4.5. Information Included When Oral Fluid Samples Are Submitted

The recently collected samples were submitted with different pieces of information related to the population of pigs from which the sample was drawn (Figure 5). The majority of the samples included basic information such as pig age and premises identification number; however, other important information such as barn or pen identification numbers, herd size, and clinical history was reported in fewer than half of the cases, making the report of more detailed clinical information less consistent.

## 4. Discussion

This study characterizes the variability of oral fluid sampling practices in the U.S. swine industry, including responses from 42 companies and clinics, representing an estimated pig population of 58 million growing pigs and 3.9 million sows. Our findings reveal the widespread and varied adoption of this sampling methodology with an important degree of variation around different procedures related to sampling across the industry, highlighting the need for standardized collection procedures to ensure consistent interpretation across production systems and to improve the reliability of detection for emerging pathogens.

In this study, 99% of respondents indicated familiarity with, or use of, oral fluids for diagnostics; however, this estimate may be inflated, as practitioners already employing this method were more likely to participate. Nonetheless, the primary objective of this study was not to measure adoption rates, but to characterize how oral fluids are being used in U.S. pig production. The primary usages of oral fluid samples were for routine, more appropriately defined as active, surveillance, and diagnostic purposes when clinical symptoms occurred (i.e., passive surveillance), which aligns with what has been described in previous studies stating that oral fluids are a practical and useful tool for diagnosis of a variety of viruses and surveillance of pig populations [2,30].

The frequency of oral fluid collection varied, mainly based on two aspects: farm type and clinical signs. In relation to farm type, GDUs are sampled more often than sow farms and growing pig farms, as samples are collected more on a regular basis than on an as-needed basis (Figure 1). Except for active screening (e.g., weekly, bi-weekly, or monthly), some veterinarians also reported that in GDU they collect oral fluids based on animal movements which refer to 2–4 days post-arrival in the GDU or 4–5 days before entry to the sow farm. This highlights the fact that oral fluid sampling is being used as a screening tool before moving and introducing new animals to other sites. In sow farms, active surveillance using oral fluids mostly happens monthly, bi-weekly, weekly, and daily, or otherwise as needed (Figure 1). For example, some veterinarians indicated that sow farms with an on-site isolation unit routinely test the gilts before allowing entry onto the sow herd, roughly 2–7 days post arrival, which agrees with the previous farm type comment regarding GDUs. On the other hand, oral fluid sample collection in growing pig farms varies. Notably, in some cases, samples were collected weekly or monthly, similar to practices in GDUs. This may reflect the animal health monitoring within multiplication flows, which differ from commercial growing pig sites. Multiplication flows produce replacement gilts for sow farms, requiring closer health surveillance to ensure herd health and minimize the risk of disease transmission to the downstream reproductive herd [31]. Clinical sign is the other factor that influenced collection frequency. Passive surveillance based on clinical signs is a common practice in growing pig farms, especially in those without active monitoring practices. We also looked at the responses reporting active surveillance using oral fluid samples in the absence of clinical signs. For these cases, when there are clinical signs, the sampling frequency increased in 64% of cases, stayed the same in 23% of cases, and decreased in 13% of responses. This represents differences in disease monitoring and intervention decisions, but generally reflects that farms monitoring oral fluids regularly intensify monitoring to track infections after the onset of clinical symptoms.

The oral fluid sample size differs based on several factors, including farm type, pen designs, and sampling purpose. From a farm-type perspective, fewer pigs per rope in the barn may be sampled at GDUs than growing pig farms, which increases the probability that every pig has access to the rope. For instance, one response reported hanging one rope per 400 grow/finishing pigs, whereas at GDU facilities, one rope is hung per pen, with each pen typically housing fewer pigs. Another respondent shared that for breeding stock, the sampling was one rope per 600 pigs, while for commercial pigs, it was one rope per barn or 1000 pigs. These examples suggested that oral fluid sample size considers the farm genetic level, as breeding stock companies closely monitor their herd health status. The number of pigs per pen seems to also influence sample size, as the number of ropes hung per pen was reported to be dependent on pen and barn design. For example, larger pens in finishing sites (e.g., 250 to 500 pigs) may use two ropes per pen instead of one. The Secure Pork Supply Plan, an animal disease preparedness initiative supported by the Pork Checkoff and USDA-APHIS also suggests that for pens with more than 25, two or more ropes may be used for oral fluid sample collection [32].

Sampling purpose also played a role, in that sampling strategies differed according to the objective. For pre-shipment testing, one respondent noted that “100% of the pens are sampled” while in weekly monitoring, “a 3-stranded rope will be divided into 3 strands that would be hung throughout the barn and then pooled together”. In isolated situations or when higher detection sensitivity is needed due to low prevalence scenarios, veterinarians may use one rope per pen instead of one rope per every two pens, to reduce the number of pigs sampled per rope. When sampling was initiated due to clinical signs, 49% of respondents maintained their active surveillance hanging strategies for the number of pigs per rope, while the same proportion (26%) either increased or decreased the number of pigs per rope in the airspace. Respondents who increased the number of pigs per rope in the airspace typically did so by reducing the number of ropes hung; for instance, when clinical signs appeared, they shifted from using one rope per pen to one rope for 1000 pigs, likely due to the fact that they were dealing with a higher prevalence scenario. Some that do not conduct active surveillance with oral fluids chose to sample 500 pigs per rope in the barn when clinical signs appeared. Conversely, respondents who reduced the number of pigs per rope in the presence of clinical signs either stopped oral fluid sampling altogether, or increased the number of ropes to achieve more accurate testing results. Therefore, the sampling strategies are influenced by detection needs, including considerations for higher sensitivity and the detection of the virus.

Although the aggregation of samples of oral fluids from different pens was not addressed in the questionnaire, 11 respondents specifically described their varied practices. One respondent reported that they “divide a 3-strand rope into three strands and place a total of six strands (two ropes) throughout the barn (~1000 pigs)”. They then “aggregate all six strands into one tube”, resulting in one oral fluid sample that represents 1000 pigs through the aggregation of six individual oral fluid samples. One respondent noted “We hang one rope per room housing approximately 300 animals. The rope is split into the three strands and hung across two pens so that all 6 pens do have samples collected in that room, then they are just pooled into one sample”, and three samples of oral fluids were aggregated into one, in this case. Aggregating samples can reduce testing costs; however, it is not recommended in previous studies and reports for oral fluid sample collection [2,32,33,34]. Henao-Diaz et al. (2020) discouraged aggregating oral fluid samples collected within barns or sites, as it may result in pathogen dilution, leading to false negatives. Instead, they suggested collecting fewer samples to confidently determine their status, rather than to pool a larger number of samples [2]. Recent studies have demonstrated that aggregating family oral fluid samples collected from a dam and her suckling piglets can effectively detect PRRSV RNA at up to 1:20 dilution and IAV at up to 1:10 dilution, both with Cycle threshold (Ct) values below 34 [35,36]. However, family oral fluid samples involve relatively fewer pigs compared to pen oral fluids from other age groups. The Secure Pork Supply Plan does not recommend aggregating oral fluid samples from ropes in different pens [32,34]. Therefore, further investigations into the cost-efficiency of different pooling strategies for different pathogens and animal ages are still necessary, highlighting the need to make informed decisions with caution when considering the pooling of oral fluid samples.

Our survey results highlight ongoing concerns and opportunities for improvement in the handling, processing, and submission of the samples of oral fluids. The written protocol for handling/processing the ropes and oral fluids was reported by 71% of respondents; however, it is concerning that some (6% of the responses) of these protocols are not accessible at the barn level. Refrigeration is recognized as critical for preserving sample quality [37], yet some samples (9%) are not refrigerated until the end of the day, which poses a significant risk to sample integrity, especially during hot summer weather in the truck. Supporting this concern, the Secure Pork Supply Plan emphasizes that oral fluids submitted on the same day should be chilled and accompanied by ice packs [32]. Moreover, previous studies have demonstrated that the half-life of RT-PCR-detectable PRRSV RNA in oral fluid samples is approximately 13 h at 30 °C [2,37], underscoring the importance of timely refrigeration. This survey data also suggested that sample submissions provide extra essential information when submitting the oral fluid samples: for example, quantitative information such as the number of pigs with clinical signs, the date on which clinical signs were first observed, the number of dead pigs, the total number of pigs in the site, and pig age. This information could potentially help within and between farm disease investigations. Of note, 17% did not include Premises ID on the submission form. This information is essential, especially as oral fluids are considered for ASF monitoring, where quickly identifying positive sites is critical. Moreover, oral fluids as a monitoring method could aid in the detection of ASF infection prior to the onset of clinical signs or sudden death, thereby supporting earlier intervention and more effective outbreak control [38,39,40].

This study has some limitations which should be taken into account when interpreting the results. First, not all answers to the open-ended questions could be quantified or systematically summarized. For instance, the original open-ended question on sample size, “What is the average number of pens sampled per barn?” was summarized in Table 2 as the number of ropes hung per barn and the number of pigs per rope in the airspace during active surveillance. However, variation in the level of detail provided limited the standardization of answers. One respondent stated “1 rope per barn but we split the rope into 3 strands for 3 pens. Pigs per barn varies.” In this case, the number of ropes hung per barn was recorded as one, while the number of pigs per rope in the airspace during active surveillance was recorded as “NA”, due to the absence of barn size information. Similarly, for the number of pens sampled per barn, three respondents indicated that they sampled all pens, and two others stated that they sampled 50% of the pens, without providing specific pen numbers. As a result, the reported number of pens sampled per barn may be underestimated in our findings. Second, this study may be subject to selection bias. The survey was distributed via email and e-letters, which was more likely to engage swine practitioners who already use oral fluid sampling. Consequently, the estimated adoption rate may be inflated, while those not utilizing oral fluids as a routine diagnostic tool may be under-represented. Nevertheless, this limitation is not expected to substantially affect the study’s primary objective, which was to characterize the use of oral fluid sampling, as current respondents are the most relevant population for capturing such practices. Third, as with all survey-based research, the representativeness of the sample is a consideration. While the 67 valid responses do not capture all swine farms in the U.S., they represent a substantial portion of the national pig population (3.9 million out of 6 million sows and 58 million out of 75 million hogs and pigs) and include geographic coverage across several major pig-producing regions. Moreover, the sample size is comparable to other U.S. swine surveys targeting veterinarians [41]. Additionally, reliance on self-reported data introduces the possibility of inaccuracies or misreporting. Some responses were overly simplistic, such as providing only a number for an open-ended question, which may slightly affect the clarity and precision of the data interpretation.

## 5. Conclusions

This study characterized current oral fluid sampling practices in the U.S. swine industry, quantifying its widespread yet variable adoption, describing its primary uses in both active and passive surveillance, and outlining the sampling protocol and the handling and submission process. Our findings reveal the varied practices of oral fluid sampling across the industry, highlighting the need for standardized procedures to ensure consistent interpretation across production systems and to improve the reliability of detection for emerging pathogens.

## Figures and Tables

**Figure 1 pathogens-14-00940-f001:**
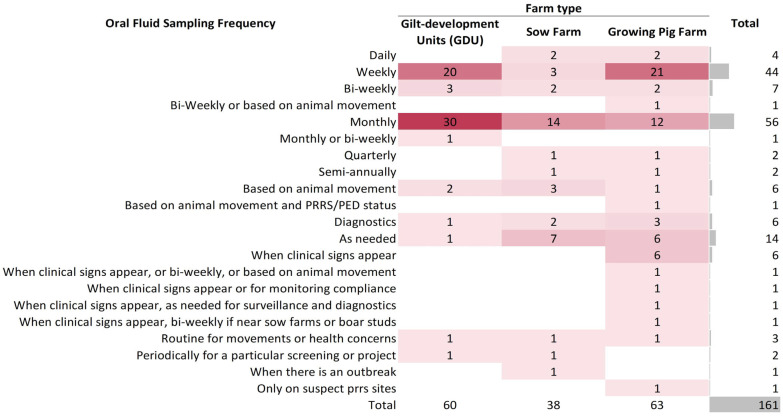
Heat map of different farm types’ oral fluid sampling frequency for active surveillance in the U.S. Darker colors indicate higher sampling frequency, while lighter colors indicate lower frequency.

**Figure 2 pathogens-14-00940-f002:**
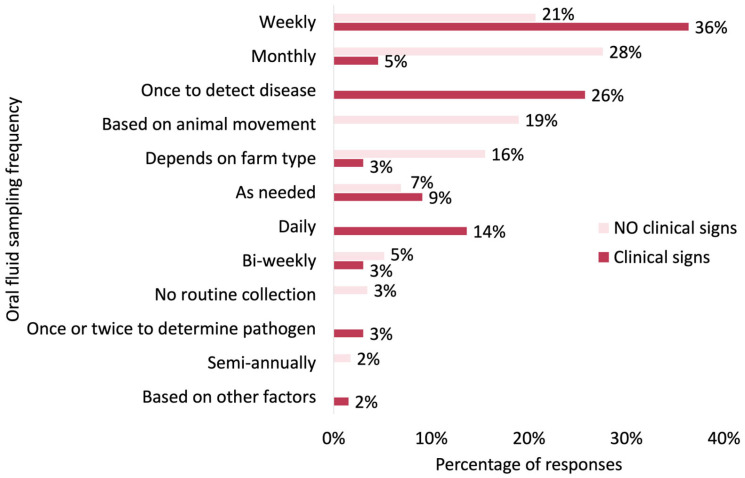
Percentage of responses on oral fluid sampling frequency according to the presence of clinical signs in the U.S. swine industry.

**Figure 3 pathogens-14-00940-f003:**
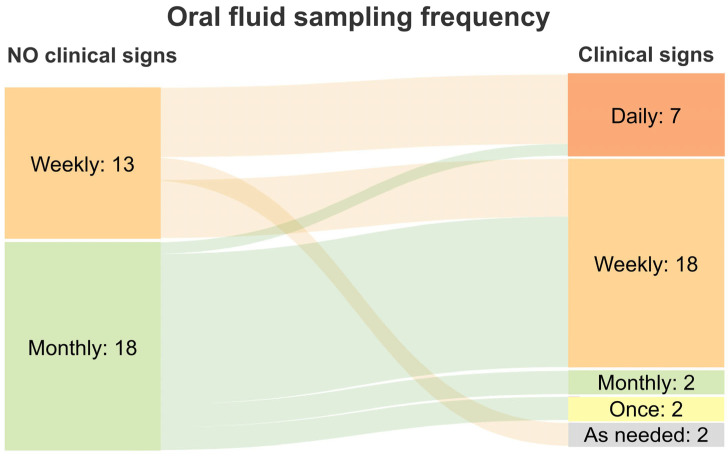
Changes in oral fluid sampling frequency in U.S. pig farms based on the presence or absence of clinical signs, according to 31 responses reporting regular monitoring (i.e., daily, weekly, monthly) for active surveillance.

**Figure 4 pathogens-14-00940-f004:**
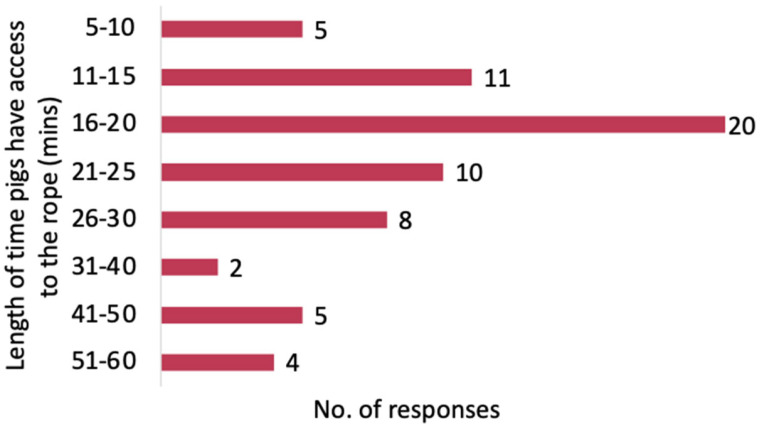
The distribution of responses on the length of time pigs have access to the rope.

**Figure 5 pathogens-14-00940-f005:**
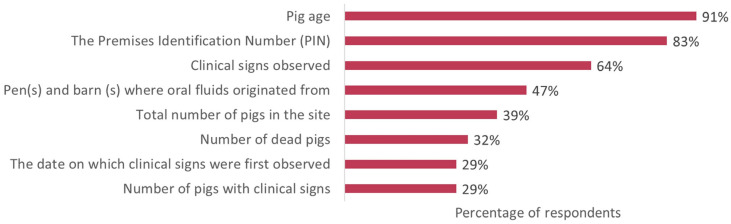
Information provided when oral fluid samples are submitted to the veterinary diagnostic laboratory.

**Table 1 pathogens-14-00940-t001:** The content structure of the online questionnaire for the characterization of oral fluid usage in pig production systems in the U.S.

Sections	Details (Corresponding Question Number in the Questionnaire *)	
Demographics	Name (Q1), contact (Q2), respondent role (Q3), affiliated companies/clinics (Q4), job title (Q5), number of sows and growing pigs they are responsible for (Q6)	
Oral fluid implementation and primary use	Use oral fluid for diagnostics (Yes/No) (Q7)	
	Primary use of oral fluids as a diagnostic sample (testing for active surveillance/passive surveillance based on clinical signs/other factors) (Q12)	
Oral fluid sampling protocols	Sampling frequency	Frequency of sampling per farm type—active surveillance (Q8)
		Frequency of sampling—no clinical signs (Q9)
		Frequency of sampling—with clinical signs (Q10)
		Other situations impact the frequency (Q11)
	Sample size	Number of ropes per barn and pigs when collecting oral fluids for active surveillance (Q13)
		Number of ropes per barn and pigs when collecting oral fluids based on clinical signs (Q14)
		How to/who determine(s) the number of ropes to hang (Q15)
		Average number of ropes used per sampled pen (Q16)
		Average number of pens sampled per barn (Q17)
		Average number of pigs represented by a single oral fluid–rope sample (Q18)
	Sampling process	Personnel collecting oral fluid samples (Q19)
		Length of time pigs have access to the rope (Q20)
		Source and the diameter of the rope (Q21)
Sample handling/processing and submission	Availability of written protocol for handling/processing the ropes (Q22)	
	Storage of collected sample (Q23)	
	Time range between collecting the oral fluid and submission to the diagnostic laboratory (Q24)	
	Destination of oral fluid samples upon collection (Q25)	
	Information includes when oral fluid samples are submitted (Q26)	

* The full content of the questionnaire is available in Appendix A.

**Table 2 pathogens-14-00940-t002:** Sample size related to oral fluid collection in pig production systems.

	Median	Min; Max	IQR (Interquartile Range)	Number of NA Answer *
Number of ropes hung per barn	2	1; 15	1; 4	5
Number of pens sampled per barn	6	1; 30	4; 10	3
Number of pigs per rope in the airspace **—active surveillance	500	20; 1700	263; 650	24
Number of pigs per rope in the airspace **—clinical sign	500	38; 2800	288; 700	22
Number of pigs represented by a single oral fluid—rope sample ***	138	5; 1200	60; 294	0

* Open-ended responses that could not be quantified are excluded from the table. The count of such answers for each question is listed here. ** Airspace was not defined in the survey; it could represent a room or a barn, depending on the layout of the farm. The number of pigs per rope in the airspace represents one rope that was hung in the barn/room of a certain number of pigs, and not all pigs in this barn/room necessarily have access to the rope. *** The number of pigs represented by a single oral fluid—rope sample is the number of pigs that the respondent believed had direct access and chewed the rope.

**Table 3 pathogens-14-00940-t003:** Descriptive summary of the average, minimum, and maximum time between oral fluid sample collection and submission to the diagnostic veterinary laboratory in the U.S.

Time Between Sample Collection and Submission	Median	Minimum	Maximum	IQR
Average (h)	24	2	48	12; 24
Minimum (h)	4.5	0	48	2; 16
Maximum (h)	36	4	96	24; 67

## Data Availability

The data are not publicly available due to privacy restrictions for survey data. Requests to access the research data should be directed to corzo@umn.edu.

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
