# Peer review of "Swine Practitioner Practices on Oral Fluid Sampling in U.S. Swine Farms: A Nationwide Survey"

_pathogens, 2025, doi:10.3390/pathogens14090940_

Round 1

Reviewer 1 Report

Comments and Suggestions for Authors

The manuscript submitted for evaluation characterize oral fluid sampling practices within the U.S. pig production systems. The authors described in detail the questionnaire used, the results obtained and also critically explained the limitations of the study to the readers. The text is well written but sometimes hard to follow, because of high volume of data, especially in results part.

The article's subject matter is not focused on pathogenic factors, but rather on a review of methods for using oral fluid. Moreover, were statistical analyses of the obtained results performed? Did the authors observe any most common procedures, as confirmed by statistical analyses?

Author Response

Dear Reviewer,

Authors would like to thank you for you insightful and constructive feedback. Below you will find our responses to your comments.

Comments 1: The manuscript submitted for evaluation characterize oral fluid sampling practices within the U.S. pig production systems. The authors described in detail the questionnaire used, the results obtained and also critically explained the limitations of the study to the readers. The text is well written but sometimes hard to follow, because of high volume of data, especially in results part.

Responses 1: We thank the reviewer for this comment. We understand the concern regarding the large volume of data presented. While the manuscript is data-rich, we believe it is organized in a way that facilitates readability, with Table 1 grouping the questionnaire into four sections and the results structured into corresponding sub-sections, allowing readers to easily follow and locate information of interest. To further improve clarity, in the revised version we have moved part of the text describing the number of answers for Question [8] from the Results to the Materials and Methods, thereby reducing textual density and helping streamline the Results section.

Comments 2: The article's subject matter is not focused on pathogenic factors, but rather on a review of methods for using oral fluid. Moreover, were statistical analyses of the obtained results performed? Did the authors observe any most common procedures, as confirmed by statistical analyses?

Responses 2: We thank the reviewer for this comment. The reviewer is correct regarding not focusing on pathogenic factors as that was not the scope of the study. The goal of the study was to characterize oral fluid usage and capture variability of implementation as it is a concern in the US industry. We believe the study provides important value by documenting current oral fluid sampling practices across U.S. swine farms and highlighting patterns that may guide the development of improved protocols. Such information is fundamental for designing surveillance systems for high-consequence pathogens, including ASF, PRRSV, and PEDV, where consistent and representative sampling is critical for timely detection. With regard to statistical analyses, our main objective was to describe the data and where possible we conducted further analyses, for instance we did perform a t-test comparing the number of pigs per rope in the airspace under routine versus clinical sampling scenarios. This analysis showed no statistically significant difference. In the revised version, we added this information on lines 312-314.

Comments 3: Figures and tables can be improved

Responses 3: We appreciate the reviewer’s suggestion regarding figures and tables. Considering the breadth and number of questions in the survey, we sought to present the results in a clear and efficient way, using diverse formats such as heatmaps and Sankey diagrams to visually summarize key findings. For submission, we ensured that all five figures were prepared at 300 dpi, maintained a consistent style across figures, and designed both figures and tables with a clean layout to maximize readability and self-explanatory value. We would be grateful if the reviewer could provide specific suggestions for improvement, which would help us refine the presentation more effectively in the revision. We would be grateful if the reviewer could provide specific suggestions for improvement so that we can address these issues more targeted and efficiently.

Reviewer 2 Report

Comments and Suggestions for Authors

Yue and co-authors present a manuscript titled Swine practitioner practices on oral fluid sampling in U.S. swine farms: A nationwide survey, that provides very interesting and valuable insights into oral fluid sampling practices in the swine industry. One of the key strengths of the study is the collection of a large number of responses, although the final analysis was based on 67 valid questionnaires. These were obtained from individuals with hands-on experience in day-to-day farm operations and represent a pig population estimated at approximately 58 million growing pigs and 3.9 million sows.

The manuscript is well written, clearly explained, and provides practical information that could serve as a useful tool for improving oral fluid sampling protocols in the swine industry. 

Specific comments:

Lines 214–222: I suggest moving this explanation to the Materials and Methods section, where it would be more appropriately placed

Lines 515–516: Could you please clarify your opinion on the potential usefulness of oral fluid sampling for ASF detection in the U.S.? Considering the nature of ASF, would mortality and clinical signs likely be detected earlier?

The inclusion of the questionnaire in the Supplementary Materials is an added value and could be useful for similar studies in other contexts or countries.

Author Response

Dear Reviewer,

Authors would like to thank you for you insightful and constructive feedback. Below you will find our responses to your comments.

Comments 1: Yue and co-authors present a manuscript titled Swine practitioner practices on oral fluid sampling in U.S. swine farms: A nationwide survey, that provides very interesting and valuable insights into oral fluid sampling practices in the swine industry. One of the key strengths of the study is the collection of a large number of responses, although the final analysis was based on 67 valid questionnaires. These were obtained from individuals with hands-on experience in day-to-day farm operations and represent a pig population estimated at approximately 58 million growing pigs and 3.9 million sows.

The manuscript is well written, clearly explained, and provides practical information that could serve as a useful tool for improving oral fluid sampling protocols in the swine industry. 

Responses 1: We sincerely thank the reviewer for the thoughtful and encouraging comments, and contributions to the manuscript.

Comments 2: Lines 214–222: I suggest moving this explanation to the Materials and Methods section, where it would be more appropriately placed

Responses 2: We thank reviewer for their comments! In the revised version, we moved this part to lines 171-179 in Materials and Methods and edited for a better fit.

Comments 3: Lines 515–516: Could you please clarify your opinion on the potential usefulness of oral fluid sampling for ASF detection in the U.S.? Considering the nature of ASF, would mortality and clinical signs likely be detected earlier?

Responses 3: We agree that it is valuable to mention the potential of oral fluids for early ASF detection. In the revised version, we added in lines 519-522 that oral fluids can support detection prior to the onset of clinical signs or sudden death, thereby contributing to more effective outbreak control.

Comments 4: The inclusion of the questionnaire in the Supplementary Materials is an added value and could be useful for similar studies in other contexts or countries.

Responses 4: We thank the reviewer for this positive comment. We are glad that the inclusion of the questionnaire in the Supplementary Materials is considered valuable and hope it can serve as a useful resource.

Reviewer 3 Report

Comments and Suggestions for Authors

Excellent paper, on a subject of considerable importance to the swine industry.

The paper describes practices used for collection of oral fluids, and represents an overview based on questions in an extensive survey. The results highlight the need for standardization of practices in order to be able to accurately understand  

Abstract: The abstract could be improved for optimal impact.  Define GDU, it is defined later in the paper, but the abstract should be a stand-alone.

Introduction: “Oral fluids are the fluids collected from the mouth by placing absorptive collectors in the oral cavity” – This is the first sentence.  For those not familiar with rope studies, it conjures an image of sticking a plastic tube in the animal’s mouth – perhaps re-word?

Discussion could be shortened slightly, but that is not mandatory.

Author Response

Dear Reviewer,

Authors would like to thank you for your insightful and constructive feedback. Below you will find our responses to your comments.

Comments 1: Excellent paper, on a subject of considerable importance to the swine industry.

The paper describes practices used for collection of oral fluids, and represents an overview based on questions in an extensive survey. The results highlight the need for standardization of practices in order to be able to accurately understand  

Abstract: The abstract could be improved for optimal impact.  Define GDU, it is defined later in the paper, but the abstract should be a stand-alone.

Responses 1: We thank the reviewer for taking the time to review this manuscript. We appreciate the encouraging and constructive feedback. We revised the beginning of the abstract in lines 11-12 to highlight the oral fluids usage; GDU was explained in the revised version in line 21.

Comments 2: Introduction: “Oral fluids are the fluids collected from the mouth by placing absorptive collectors in the oral cavity” – This is the first sentence.  For those not familiar with rope studies, it conjures an image of sticking a plastic tube in the animal’s mouth – perhaps re-word?

Responses 2: Revised in lines 31-35.

Comments 3: Discussion could be shortened slightly, but that is not mandatory.

Responses 3: We thank the reviewer for this suggestion. We reviewed the Discussion and made minor edits to improve clarity and conciseness, changes are made on lines 400-401, lines 408-410. While the Discussion contains several descriptions and examples of survey answers, we believe these are necessary to retain, as they illustrate the variability in practices and responses. Including such examples helps readers better interpret the survey findings and understand the context behind the reported results.

Round 2

Reviewer 1 Report

Comments and Suggestions for Authors

Dear Authors,

Thank you for your answers and comprehensive explanations. I agree with the authors that standardization of oral fluid collection methods is needed to make diagnostics more effective.

I had some concerns about image 3, which, despite its visual appeal, was somewhat difficult for me to understand. However, after further consideration, I think it should remain in the manuscript.

The only issue that needs improvement, in my opinion, is the inclusion of question number 8 in [] (e.g. in the line 172), which may suggest citing article number 8.

Author Response

Comments 1: Thank you for your answers and comprehensive explanations. I agree with the authors that standardization of oral fluid collection methods is needed to make diagnostics more effective.

I had some concerns about image 3, which, despite its visual appeal, was somewhat difficult for me to understand. However, after further consideration, I think it should remain in the manuscript.

Responses 1: We sincerely thank the reviewer for their careful eye and detailed feedback. We appreciate your support in retaining Figure 3 in the manuscript. To improve its clarity, we have explicitly noted in the revised version (line 264) that this is a Sankey diagram.

Comments 2: The only issue that needs improvement, in my opinion, is the inclusion of question number 8 in [] (e.g. in the line 172), which may suggest citing article number 8.

Responses 2: We thank the reviewer for catching this! We have revised Table 1 by changing the format from “[x]” to “(Qx)” to avoid confusion with reference citations. We have also screened the manuscript and made corresponding adjustments in the text (lines 172 and 222).

All these changes are highlighted in green in the revised manuscript.